:ᑕ̣: PLOS | ONE

# TSG-6 in extracellular vesicles from canine mesenchymal stem/stromal is a major factor in relieving DSS-induced colitis

Ju-Hyun An[1], Qiang Li[1¤], Min-Ok Ryu[1], A-Ryung Nam[1], Dong-Ha Bhang[2], Yun-Chan Jung[3], Woo-Jin Song[1☉]*, Hwa-Young Youn[1☉]*

**1** Laboratory of Veterinary Internal Medicine, Department of Veterinary Clinical Science, College of Veterinary Medicine, Seoul National University, Seoul, Republic of Korea, **2** Department of Molecular Cell Biology, Samsung Biomedical Research Institute, Sungkyunkwan University School of Medicine, Suwon-si, Gyeonggi-do, Republic of Korea, **3** Chaon Corporation, Bundang-gu, Seongnam-si, Gyeonggi-do, Republic of Korea

☉ These authors contributed equally to this work.
¤ Current Address: Department of Veterinary Medicine, College of Agriculture, YanBian University, YanJi, JiLin, China
* woogin1988@snu.ac.kr (WJS); hyyoun@snu.ac.kr (HYY)

**Data Availability Statement:** All relevant data are within the manuscript and its Supporting Information files.

## Abstract

Adipose tissue derived mesenchymal stem/stromal cell (ASC)-derived extracellular vesicles (EV) have been reported to be beneficial against dextran sulfate sodium (DSS)-induced colitis in mice. However, the underlying mechanisms have not been fully elucidated. We hypothesize that the tumor necrosis factor-α-stimulated gene/protein 6 (TSG-6) in EVs is a key factor influencing the alleviation of colitis symptoms. DSS-induced colitis mice (C57BL/6, male, Naïve = 6, Sham = 8, PBS = 8 EV = 8, CTL-EV = 8, TSG-6 depleted EV = 8) were intraperitoneally administered EVs (100 ug/mice) on day 1, 3, and 5; colon tissues were collected on day 10 for histopathological, RT-qPCR, western blot and immunofluorescence analyses. In mice injected with EV, inflammation was alleviated. Indeed, EVs regulated the levels of pro- and anti-inflammatory cytokines, such as TNF-α, IL-1β, IFN-γ, IL-6, and IL-10 in inflamed colons. However, when injected with TSG-6 depleted EV, the degree of inflammatory relief was reduced. Furthermore, TSG-6 in EVs plays a key role in increasing regulatory T cells (Tregs) and polarizing macrophage from M1 to M2 in the colon. In conclusion, this study shows that TSG-6 in EVs is a major factor in the relief of DSS-induced colitis, by increasing the number of Tregs and macrophage polarization from M1 to M2 in the colon.

## Introduction

Inflammatory bowel disease (IBD) is a chronic debilitating disease that affects both humans and dogs, characterized by abdominal pain and diarrhoea. It may result in significant morbidity and mortality, with compromised quality of life and life expectancy. Clinical signs may be controlled by single or combination therapy, including dietary modifications, antibiotics and immune-suppressants. However, since there is no clear treatment method, clinical recurrence frequently occurs even after treatment, and thus, new therapeutic agents need to be sought [1].

**Funding:** This study received support from the Research Institute for Veterinary Science, Seoul National University and Basic Science Research Program of the National Research Foundation of Korea. Author YCJ receives support in the form of salary from the commercial company the Chaon Corporation. The funders did not have any additional role in the study design, data collection and analysis, decision to publish, or preparation of the manuscript. The specific roles of YCJ are articulated in the 'author contributions' section.

**Competing interests:** Author YCJ is employed by the commercial company the Chaon Corporation. There are no patents, products in development, or marketed products to declare. This does not alter the authors' adherence to all PLOS ONE policies on sharing data and materials. The authors declare that no other competing interests exist.

**Abbreviations:** ANOVA, Analysis of variance; BCA, Bicinchoninic acid; cASC, Canine adipose tissue-derived mesenchymal stem/stromal cell; CD, Cluster of differentiation; Con A, Concanavalin A; CTL, Control; cPBMC, Canine peripheral blood mononuclear cell; DAI, Disease-activity index; DMEM, Dulbecco's modified Eagle's medium; DMSO, Dimethyl sulfoxide; DPBS, Dulbecco's phosphate-buffered saline; DSS, Dextran sulfate sodium; ELISA, Enzyme-linked immunosorbent assay; EV, Extracellular vesicle; FBS, Fetal bovine serum; FITC, Fluorescein isothiocyanate; FOXP3, Forkhead box P3; GAPDH, Glyceraldehyde 3-phosphate dehydrogenase; H&E, Hematoxylin and eosin; IBD, Inflammatory bowel disease; IFN, Interferon; IL, Interleukin; IP, Intraperitoneal; LPS, Lipopolysaccharides; M1, Macrophage M1 type; M2, Macrophage M2 type; NO, Nitric oxide; PBMC, Peripheral blood mononuclear cell; PBS, Phosphate-buffered saline; PE, Phycoerythrin; RBC, Red blood cell; RT-qPCR, Reverse transcription quantitative polymerase chain reaction; siRNA, Small-interfering RNA; siTSG-6, Small-interfering TSG-6; TEM, Transmission Electron Microscope; TGF-β, Transforming growth factor-β; TNF-α, Tumor necrosis factor-α; Treg, Regulatory T cell; TSG-6, Tumor necrosis factor-α-stimulated gene/protein-6.

Adipose tissue derived mesenchymal stem/stromal cells (ASCs) are of great interest as novel therapeutics for IBD patients because of their unique ability to regulate immune cells and heal damaged colonic tissue [2]. Particularly, it has been found that extracellular vesicles (EVs, 40–1000 nm sized circular membrane fragments shed from the cell surface) secreted from cells can mediate the delivery of secreted molecules in cell-to-cell communication. Thus, studies on the use of EVs as an alternative to stem cells have been actively conducted [3, 4]. Recently, various studies have been carried out on the application of EVs as therapeutic agents in various pre-clinical models such as acute kidney injury, hepatitis, cystitis and uveitis[5–8]. In addition, injecting EVs into DSS-induced colitis mouse models has shown that not only does it improve activity and appetite, but it also alleviates inflammation in the colon [9]. Although these studies reported that damaged tissues were improved following treatment with EVs, the factors responsible for the protective effects have yet to be elucidated. If stem cell-derived EVs are to be used as therapeutic agents in the future, in-depth mechanistic studies to determine which factors are most highly associated with the ability of EVs to alleviate inflammation must be conducted.

Tumor necrosis factor (TNF)-α stimulated gene/ protein 6 (TSG-6) secreted from stem cells is a major factor responsible for regulation of inflammatory responses [10–12]. Moreover, several studies have shown that TSG-6 plays important roles in attenuating DSS-induced colitis in mice by altering the composition of immune cells in the colon [11, 13]. However, studies on TSG-6 in EVs have not yet been conducted, and, thus, further studies are required.

In particular, IBD is related to an immunological imbalance in the intestinal mucosa, which is primarily associated with cells of the adaptive immune system that respond to self-antigens produced under inflammatory conditions in such patients [14, 15]. Among the intestinal immune cells, regulatory T cells (Tregs) control the balance of immune cell functions and play critical roles in self-tolerance and homeostasis in the colon [16]. Additionally, macrophages of the colon are essential for local homeostasis and play an important role in inflammation and protective immunity[17]. And they are classically divided into two major types: A very basic dichotomous view classified M1 as having an inflammatory phenotype, while M2 was considered an anti-inflammatory macrophage[18]. For this reason, various methods of increasing Tregs and macrophage M2 types in the colon in colitis experimental models have been proposed as treatment options, with stem cell derived EV being one of them[9, 19]. However, there is still a lack of research on mechanisms, and there is a need for further study.

Therefore, this study focused on elucidating the role of TSG-6 in EVs in mitigating colitis and as well as to describe potential mechanisms responsible for any protective effects observed in DSS-induced mouse models of colitis. We also investigated the effect that EVs have on Treg and M2 within the colon and how TSG-6 in EVs affects the Treg and M2 population in inflamed colons.

## Materials and methods

All animal experimental procedures were approved by the Institutional Animal Care and Use Committee of Seoul National University (SNU), Republic of Korea, and all protocols were in accordance with approved guidelines (SNU; protocol no. SNU-180829-2-1).

### Isolation, culture, and characterization of cASCs

The adipose tissue was obtained from a healthy adult female dog during ovariohysterectomy at the SNU Veterinary Medicine Teaching Hospital, with the owner's consent. ASCs were isolated from tissues and cultures as previous described [11, 20]. The cells were characterized for the expression of several stem cell markers by flow cytometry before they were used in the

experiments. Additionally, the differentiation ability of cells was confirmed for cASCs at passages 3 and 4, and these cells were used in subsequent experiments. The methods used for isolating, culturing, and characterizing stem cells are described in detail in S1 File, and corresponding results were described in detail in S1 Fig. This experiment was repeated a minimum of three times to confirm reproducibility.

## Small interfering RNA (siRNA) transfection of cASCs

To obtained TSG-6 depleted EV, when cASCs reached approximately 70% confluence, they were transfected for 48 h with TSG-6 siRNA or control siRNA (sc-39819 and sc-27007, respectively, Santa Cruz Biotechnology, Dallas, TX, USA) using Lipofectamine RNAiMAX (Invitrogen, Carlsbad, CA, USA) according to the manufacturer's instructions [11, 20, 21]. TSG-6 knockdown was confirmed by gel PCR and RT-qPCR. The cells were washed twice with PBS, and the media was exchanged with DMEM containing exosome-depleted FBS for an additional 48 h before collecting TSG-6-depleted EVs. EVs were obtained using an ultracentrifuge as above described, and the relative TSG-6 protein levels in EVs were measured by western blot analysis. Protein concentrations were determined by performing BCA assays. The total protein content (20 μg) of each sample was subjected to sodium dodecyl sulfate-polyacrylamide gel electrophoresis (SDS-PAGE) and immunoblotting with antibodies against CD63 (NBP2-42225; Novus Inc., Littleton, CO, USA)) and TSG-6 (sc-30140; Santa Cruz Biotechnology). These results were described in S2 Fig. This experiment was repeated a minimum of three times to confirm reproducibility.

## Isolation and characterization of EVs

cASCs were cultured for 48 h in Dulbecco's Modified Eagle's Medium (DMEM; PAN-Biotech, Aidenbach, Germany) supplemented with 10% Exosome-depleted Fetal bovine serum (FBS; Systembio, CA, USA) and 1% penicillin-streptomycin (PS; PAN-Biotech). The supernatant from each cultured cASC sample was collected on ice and centrifuged at $300 \times g$ for 10 min to remove the cells. Each supernatant was transferred to a fresh tube, centrifuged at $2000 \times g$ for 30 min to remove cellular debris, and then passed through a 0.22-μm filter (Millipore, Billerica, MA, USA) to remove the large vesicles. Each supernatant was transferred to a fresh tube and centrifuged at $110,000 \times g$ (Beckman Avanti Centrifuge J-26XP with 70Ti rotor, Brea, CA, USA) for 80 min, washed with Dulbecco's phosphate-buffered saline (DPBS), and purified by centrifugation at $110,000 \times g$ for 80 min. All centrifugation steps were performed at 4˚C. Each pellet was resuspended in DPBS and sterilized by filtration through a 0.22-μm filter (Fig 1C). The total protein concentration in each EV preparation was quantified by performing bicinchoninic acid (BCA) assays, and the samples were stored at −80˚C until use. Protein markers of purified EVs were determined by western blotting with antibodies against CD63 (Novus Inc.) and CD9 (GTX76185; GeneTex, Irivine, CA, USA). The EV morphology was characterized by transmission electron microscopy. Briefly, 10 μL of an EV suspension was placed on clean parafilm. A 300-mesh formvar/carbon-coated electron microscopy grid was floated on the drop, with the coated side facing the suspension, and allowed to adsorb for 20 min at 23 ± 2˚C. The grid was transferred to a 100 μL drop of distilled water and left to stand for 2 min. The grid was then transferred to a 50 μL drop of 2% uranyl acetate for negative staining for 10 min, followed by observation under a TEM (LIBRA 120, Carl Zeiss, Germany) at 120 kV. The size distribution of the particles was measured using a Zeta-potential & Particle size Analyzer (ELSZ-1000ZS, Otsuka Electronics, Osaka, Japan). This experiment was repeated a minimum of three times to confirm reproducibility.

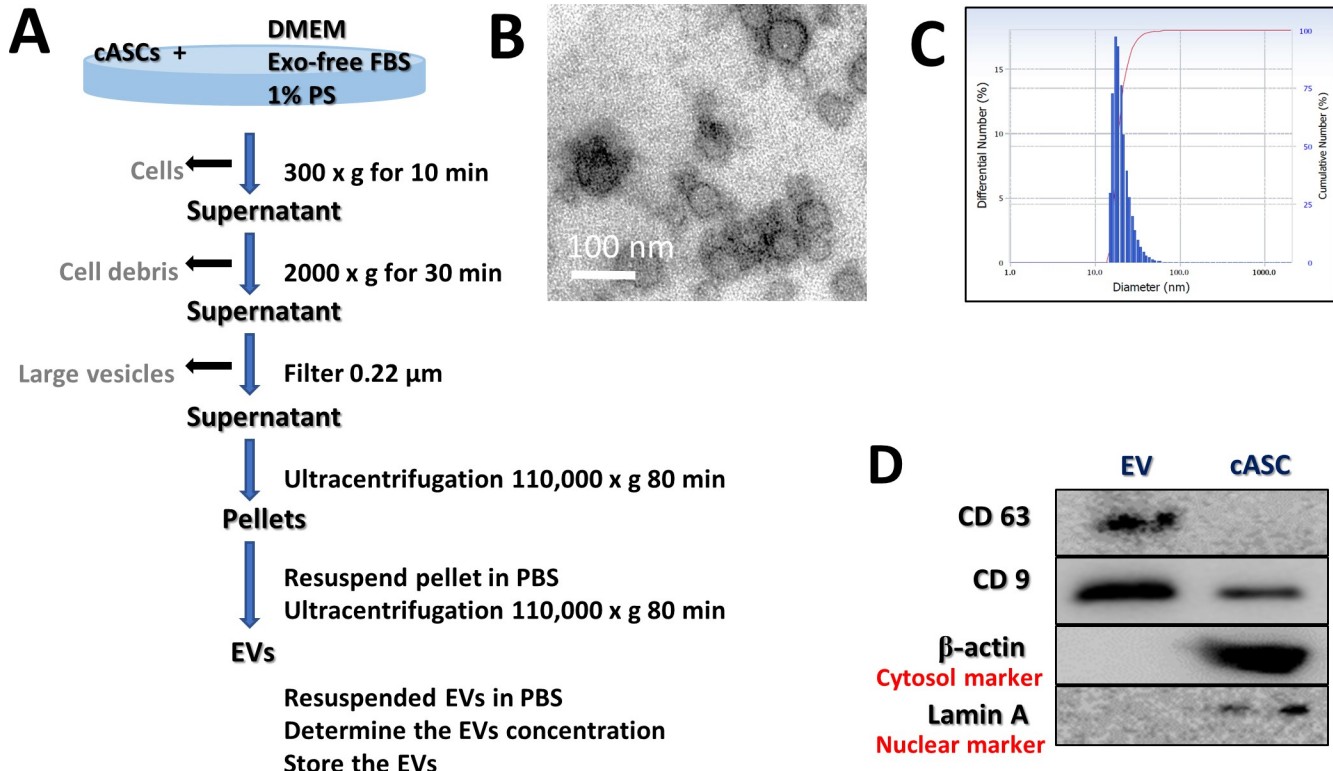

**Fig 1. Characterization of cASC-EVs.** (A) Schematic overview of the UC-based purification protocol (B) Scanning electron microscopy micrographs of cASC-EV showed spheroid shaped vesicles with diameters of approximately 20–100 nm. Scale bar, 50 nm. (C) Size-distribution analysis of purified cASC-EVs showed that the vesicle diameters were approximately 20–100 nm. (D) Immunoblotting analysis of common EV markers, where 10 μg of total protein was loaded in each lane. cASCs-EVs expressed CD63 and CD9, while beta actin and lamin A showed lower expression. The displayed data represent at least three repeated experiments with consistent results.

## Isolation of peripheral blood mononuclear cells (PBMCs) and non-adherent cells

Using citrate phosphate dextrose adenine-containing tubes, we collected blood samples (30 mL) from three healthy dogs. The blood samples were diluted with an equal volume of phosphate-buffered saline (PBS) and then layered over Ficoll-Plaque PLUS (GE Healthcare Life Sciences, Little Chalfont, UK) in a conical tube. After centrifugation at $400 \times g$ for 30 min, the buffy coat layer was carefully collected. The collected samples were incubated with red blood cell-lysis buffer at room temperature for 10 min. After adding PBS, each sample was centrifuged at $400 \times g$ for 10 min, and the washing and centrifugation steps were repeated. Canine PBMCs (cPBMCs) were resuspended in Roswell Park Memorial Institute (RPMI) medium (Pan-Biotech, Dorset, Germany) supplemented with 10% EV-free FBS and 1% PS. Non-adherent cells were obtained after 24 hours.

## Co-culture experiments

DH82 cells, a canine macrophage-like cell line, were purchased from the Korean Cell Line Bank (Seoul, Korea). DH82 cells were seeded in 6-well plates ($1\times10^{6}$ cells/well), then incubated for 24 h. After adherence to the plates was confirmed, the DH82 cells were treated with LPS (200 ng/mL; Sigma-Aldrich) or control for 24 h. Similarly, canine lymphocytes were seeded in

6-well plates ($1\times10^6$ cells/well) and exposed to Con A (5 μg/mL) or control for 24 h. Next, the medium was removed and replaced with media containing EV (100 μg/well) derived from naïve, si RNA and si TSG-6 cASCs. Next, the cells were incubated for 48 h and then harvested for RNA extraction and flow cytometry analysis.

## RNA extraction, cDNA synthesis, and reverse transcription-quantitative polymerase chain reaction

RNA was extracted using the Easy-BLUE Total RNA Extraction Kit (Intron Biotechnology). Next, cDNA was synthesized using LaboPass M-MuLV Reverse Transcriptase (Cosmogenetech, Seoul, Korea), according to the manufacturer's instructions. cDNA samples were assayed using AMPIGENE® qPCR Green Mix Hi-ROX with SYBR Green Dye (Enzo Life Sciences, Farmingdale, NY, USA), according to the manufacturer's instructions. Expression levels were normalized to those of glyceraldehyde 3-phosphate dehydrogenase (*GAPDH*). The sequences of the primers used in this experiment are shown in Table 1.

## Flow cytometry analysis

To evaluate Treg polarization, PBMC-derived lymphocytes cocultured with cASC-EVs were harvested. Obtained cells ($1 \times 10^6$) were suspended in 100μL DPBS and 1 μL of each primary antibody against the following proteins: FOXP3-PE (eBioscience, San Diego, CA, USA; 1:100), CD3-FITC (MCA1774F; Bio-Rad, San Diego, CA, USA; 1:100), CD206-FITC (eBioscience, San Diego, CA, USA; 1:100) and CD11c-PE (eBioscience, San Diego, CA, USA; 1:100). After incubation for 1 h at 23 ± 2˚C, the cells were washed with DPBS. Unstained cells were used as controls for autofluorescence. Cell fluorescence was analyzed with a flow cytometer (FACS Aria II; BD bioscience). The results were analyzed using FlowJo 7.6.5 software (Tree Star, Inc., Ashland, OR, USA).

## ELISA analysis

The protein levels of interleukin 10 (IL-10) in each cell culture supernatant was detected using an IL-10 Enzyme-Linked Immunosorbent Assay Kit (ELISA; eBioscience), according to the manufacturer's instructions.

## Mice

To determine the therapeutic effect of ASC-derived EVs on colitis, we induced mouse colitis with DSS (36–50 kDa; MP Biomedical, Solon, OH, USA). Male C57BL/6 mice (6 to 8-week-old and weighing 18 to 20 g) were purchased from Nara Bio (Gyeonggi, Korea) and acclimatized for 7 days with a 12 h light/dark cycle at 22˚C and 60% humidity before performing the experiments. For environmental enrichment, 3 to 4 mice were raised in polycarbonate cages ($324 \times 221.5 \times 130$ mm) containing clean bedding (shavings; Nara Biotech), cardboard boxes, and tunnels. Mice were fed a standard laboratory rodent diet (Central Lab Animal Inc., Seoul, Korea) and water *ad libitum*. At the start of the experiments, the health status of the mice was evaluated by measuring their weight, vitality, and defecation; the experiments were conducted on mice with no abnormal symptoms. The mice were administered 3% DSS in their drinking water from days 0 to 7. The studies were conducted using 46 animals, and the mice were randomly divided into 6 groups with 6–8 mice per group (naïve = 6, Sham = 8, PBS = 8, EV = 8, CTL-EV = 8, TSG-6 depleted EV = 8). The mice were treated with 100 μL DPBS, with or without cASC-EV (100 μg/mouse), by intraperitoneal (IP) injection on days 1, 3 and 5. The disease-activity index (DAI) represents the combined score of weight loss relative to the initial

**Table 1. Sequences of PCR primers used in this study.**

| Gene | Forward (5'-3') | Reverse (5'-3') | Reference |
|---|---|---|---|
| cGAPDH | TTAACTCTGGCAAAGTGGATATTGT | GAATCATACTGGAACATGTACACCA | [2] |
| cTNF-α | TCATCTTCTCGAACCCCAAG | ACCCATCTGACGGCACTATC | [2] |
| cIFN-γ | TTCAGCTTTGCGTGATTTG | CTGCAGATCGTTCACAGGAA | [22] |
| cIL-1β | AGTTGCAAGTCTCCCACCAG | TATCCGCATCTGTTTTGCAG | [23] |
| cIL-6 | ATGATCCACTTCAAATAGTCTACC | AGATGTAGGTTATTTTCTGCCAGTG | [2] |
| cIL-10 | ATTTCTGCCCTGTGAGAATAAGAG | TGTAGTTGATGAAGATGTCAAGCTA | [2] |
| cTSG-6 | TCCGTCTTAATAGGAGTGAAAGATG | AGATTTAAAAATTCGCTTTGGATCT | [2] |
| cCD4 | TGCTCCCAGCGGTCACTCCT | GCCCTTGCAGCAGGCGGATA | [24] |
| cCD25 | GGCAGCTTATCCCACGTGCCAG | ATGGGCGGCGTTTGGCTCTG | [24] |
| cFOXP3 | AAACAGCACATTCCCAGAGTTC | AGGATGGCCCAGCGGATCAG | [25] |
| ciNOS | GAGATCAATGTCGCTGTACTCC | TGATGGTCACATTTTGCTTCTG | [23] |
| cCD206 | GGAAATATGTAAACAGGAATGATGC | TCCATCCAAATAAACTTTTTATCCA | [23] |
| mGAPDH | AGTATGTCGTGGAGTCTACTGGTGT | AGTGAGTTGTCATATTTCTCGTGGT | [14] |
| mTNF-α | CCAGGAGAAAGTCAGCCTCCT | TCATACCAGGGCTTGAGCTCA | [14] |
| mIFN-γ | GATGCATTCATGAGTATTGCCAAGT | GTGGACCACTCGGATGAGCTC | [14] |
| mIL-1β | CACCTCTCAAGCAGAGCACAG | GGGTTCCATGGTGAAGTCAA | [14] |
| mIL-6 | TCCAGTTGCCTTCTTGGGAC | GTACTCCAGAAGACCAGAGG | [14] |
| mIL-10 | TGGCCCAGAAATCAAGGAGC | CAGCAGACTCAATACACACT | [14] |
| mCD4 | GAGAGTCAGCGGAGTTCTC | CTCACAGGTCAAAGTATTGTTG | [26] |
| mCD25 | CTCCCATGACAAATCGAGAAAGC | ACTCTGTCCTTCCACGAAATGAT | [26] |
| mFOXP3 | TTGGCCAGCGCCATCTT | TGCCTCCTCCAGAGAGAAGTG | [26] |
| miNOS | AAAGGAAATAGAAACAACAGGAACC | GCATAAAGTATGTGTCTGCAGATGT | [27] |
| mCD206 | AACGGAATGATTGTGTAGTTCTAGC | TACAGGATCAATAATTTTTGGCATT | [28] |
| mArg | CAGAAGAATGGAAGAGTCAG | CAGATATGCAGGGAGTCACC | [29] |

body weight (grades 0–4; 0, no weight loss; 1, < 10% loss; 2, 10–20% loss; 3, 20–30% loss; and 4, 30–40% loss), stool consistency (grades 0–2; 0, normal; 1, soft; and 2, liquid), the presence of blood in the feces and anus (grades 0–2; 0, negative fecal occult bleeding; 1; positive fecal occult bleeding; and 3, visible fecal occult bleeding), and general activity (grades 0–2; 0, normal; 1, mildly to moderate depressed and 2, severely depressed). The DAI score of colitis was calculated independently by two blinded investigators. The score for each parameter was summed from day 0 to the day of sacrifice, and the summed score was averaged to yield the final score. Since this model has been verified in several studies, it is unlikely to be accompanied by unexpected pain. However, should three or more of the following abnormal behaviors be observed: behaviorally excessive waist bending, self-cutting, aggressive behavior, stabbing of other mice, injury due to fall, failure to build a nest, abnormally rough hair, abnormal posture, or convulsions, the animal was euthanized within one day and the experiment terminated. On day 10 of the study, the veterinarian euthanized all mice with xylazine infusion and CO2 inhalation according to the approved institutional animal ethics protocol.

## Histological evaluation

Colon tissues were fixed in 10% formaldehyde for 48 h, embedded in paraffin, cut into 4-μm sections, and stained with hematoxylin and eosin (H&E). Histological scores are provided in Table 2. Because DSS-related injury varies, two slides from each colon section were assessed per mouse, and at least three areas on each slide were examined.

**Table 2. Histological assessment.**

| Score | Mucosal thickness & hyperplasia | Inflammatory cell extent | Damaged to the crypt |
|---|---|---|---|
| 0 | Normal | Normal | An intact crypt |
| 1 | Minimal | 11–25% of mucosa | Loss of the basal 1/3 of crypt |
| 2 | Mild | 26–50% of mucosa | Loss of basal 2/3 of crypt |
| 3 | Moderate | Mucosa and submucosa | Entire loss of crypt |

### Immunofluorescence analysis

Colon sections were deparaffinized and rehydrated, and antigen retrieval was carried out in 10 mM citrate buffer. Sections were then washed and blocked with blocking buffer containing 1% bovine serum albumin and 0.1% tween 20 for 30 min. The sections were then incubated overnight at 4˚C with mouse monoclonal anti-Forkhead box (Fox) P3 (1:100; Santa Cruz Biotechnology) and mouse monoclonal anti CD206 (1:100; Santa Cruz Biotechnology). The colon sections were washed three times with DPBS. Then, the sections were incubated FITC conjugated anti mouse (1:500; Santa Cruz) for 1 hr. After that, they were washed three times with DPBS. All samples were mounted using Vectashield mounting medium containing 4',6-diamidino-2-phenylindole (DAPI; Vector Laboratories, Burlingame, CA, USA). The samples were observed using an EVOS FL microscope (Life Technologies, Darmstadt, Germany). Immuno-reactive cells were counted in 20 random fields per group, and the percentage of CD206[+] positive cells and FOXP3[+] positive cells was calculated in colon sections from the same mice.

### Statistical analysis

GraphPad prism (version 6.01) software (GraphPad, Inc., La Jolla, CA, USA) was used for statistical analysis. the differences between two groups were analyzed using Student's t-tests and differences between more than two groups were analyzed using one-way analysis of variance (ANOVA) followed by Bonferroni multiple comparison test. The results are presented as the mean value ± standard deviation (SD). Differences with a value of $P < 0.05$ were considered as statistically significant.

## Results

### Characterization of cASC-EV

The characterization of stem cells used for these studies are described in detail in S1 Fig. The EVs were separated from stem cell culture media by ultracentrifugation. Schematic overview describing this protocol is presented in Fig 1A. Approximately 100 μg of EVs was produced in the media in which $1 \times 10^6$ cells were seeded and grown for 2 days. Electron microscopic analysis demonstrated that the EVs were round-shaped and 50–100 nm in diameter (Fig 1B). Using a particle-size analyzer, the EVs were confirmed to be less than 100 nm in diameter (Fig 1C). In addition, positive markers of EVs such as CD63 and CD9 were identified by western blotting, whereas negative markers of EVs such as Lamin A (a nuclear marker) and beta actin (a cytosolic marker) were present in lower abundance (Fig 1D). Whole-cell lysates were used as a positive control. Our findings suggest that the EVs contained little or no cellular matrix and nuclei, which are intracellular components.

## Production of EVs containing less TSG-6 from cASCs

To reduce TSG-6 in EVs, cASCs were transfected with si-TSG-6. No differences were observed in the cell viability and cell differentiation potentials between transfected and untreated stem cells. Furthermore, the size and shape of the EVs secreted from the transfected stem cells did not differ from those of the untreated stem cells, and no difference occurred in the amount of EVs produced (S1 Fig). Further, the TSG-6 mRNA levels in the transfected cASCs were reduced by over 50% (S2 Fig) while the TSG-6 protein levels in the EVs were reduced to less than half of those in the naïve and control siRNA-treated groups (S2 Fig)).

## IP administration of cASC-EVs containing TSG-6 played a crucial role in alleviating IBD

In this study, all animals met the euthanasia criteria before being sacrificed. Injecting EV into mice that did not induce colitis (sham group) showed no difference in vitality, weight, and stool consistency from naive mice. In addition, the sham group was not different from the naive group in the colon length and histological examination. In the DSS-administered group, significant weight loss and clinical indices including DAI which is based on body weight (Fig 2A), stool consistency, bloody diarrhea and general activity were found to worsen compared to the healthy group; while the EV group exhibited improved weight and DAI compared to the DSS group (Fig 2B). Moreover, shortening of the colon length significantly improved in the EV group compared with that in the PBS-treated group. However, in TSG-6 depleted EV group was found to have a shorter colon length than the EV and CTL-EV groups (Fig 2C). The EV group showed greatly decreased histological colitis scores for mucosal thickness, mucosal hyperplasia, extent of inflammation, and crypt damage. In addition, the anti-inflammatory effect of CTL-EV was similar to that of naive-EV, whereas the effect of TSG-6-depleted EV was insignificantly decreased. (Fig 2D).

## TSG-6 in cASC-EV modulate pro- and anti- inflammatory cytokine inflamed colon

The colon of EV-treated mice showed reduced levels of inflammatory cytokines (TNF-α, IFN-γ, IL-1β, IL-6 and iNOS) and elevated levels of an anti-inflammatory/regulatory cytokine (IL-10), compared to that of untreated mice with colitis and TSG-6-depleted EV-treated mice (Fig 3).

## TSG-6 was a major factor in increasing Tregs in Con A stimulated lymphocytes in vitro

We found that CD4, CD25 mRNA expression levels increased in the EV group, but decreased in the TSG-6 depleted EV group (Fig 4A). Additionally, the protein levels of IL-10 were measured in the lymphocyte cultured medium. The EV group showed increased IL-10 expression, whereas the TSG-6-depleted EV group showed decreased IL-10 expression (Fig 4B). In addition, to determine whether the increased number of Tregs among total T cells was associated with TSG-6 in the EVs, the degree of Treg activation was confirmed by fluorescence-activated cell sorting. Accordingly, the number of FOXP+ cells among CD3+ cells increased in the EV group. However, in the TSG-6-depleted EV group, the proportion of FOXP+ cells decreased (Figs 4C and S5). These results demonstrate that the immunomodulatory effects of EV were related to TSG-6.

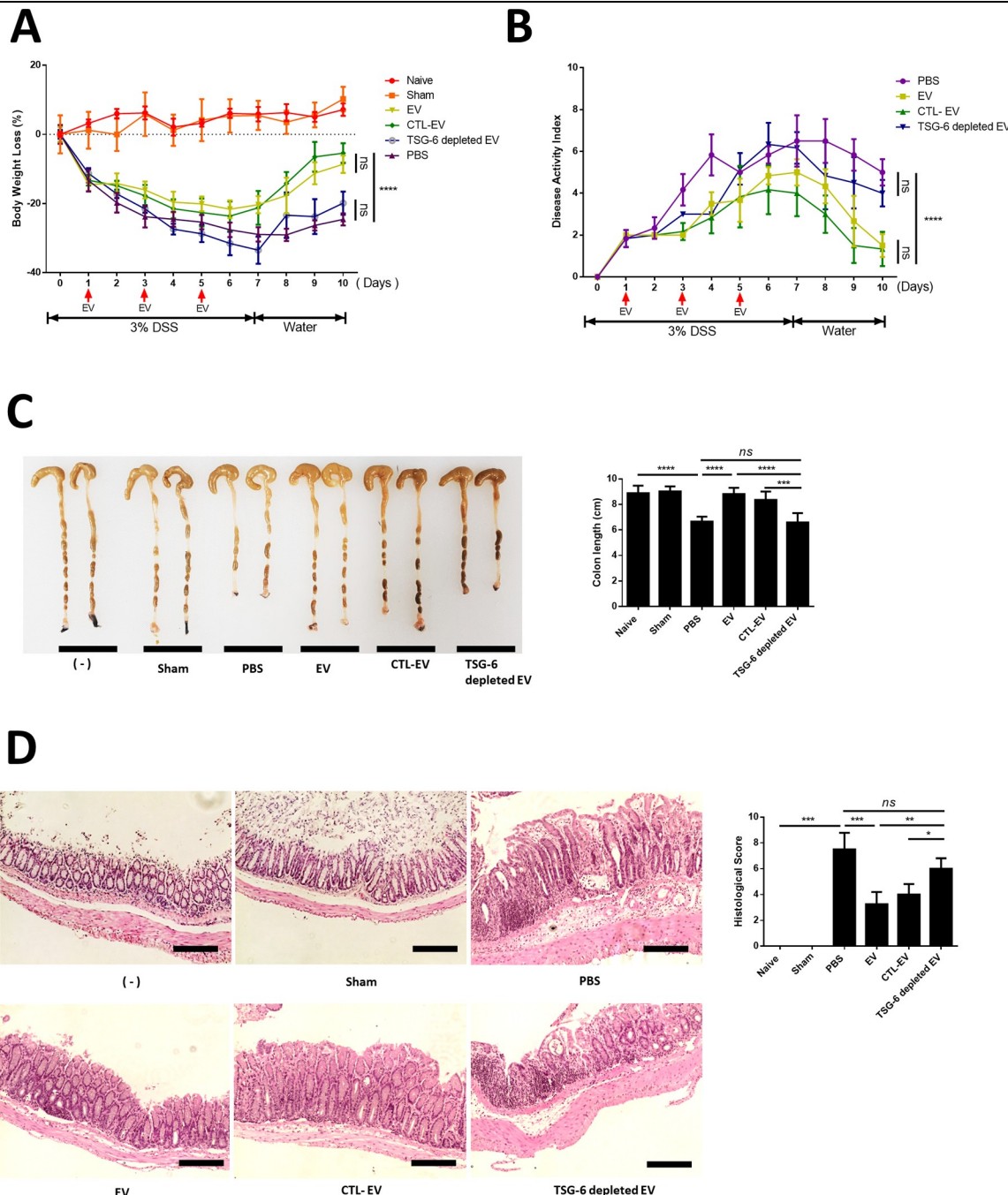

**Fig 2. cASC-EV injection ameliorated DSS-induced colitis in mice.** EVs (100 μg), TSG-6 depleted EVs (100 μg), control EVs (100 μg), or vehicle control were injected IP one day after mice were administered 3% DSS. On days 3 and 5, the mice in each group were re-injected with EVs (100 μg), TSG-6 depleted EVs (100 μg), control EVs (100 μg), or vehicle control (PBS). Mice were monitored for changes in (A) body weight, (B) DAIs, and (C) colon lengths. (D) H&E staining of colon sections and histological scores are shown. Scale bars, 100 μm. The results are shown as mean ± standard deviation (n = 6–8 in each group, *P < 0.05, **P < 0.01, ***P < 0.001, ****P < 0.0001, as determined by one-way ANOVA)

## TSG-6 in EV is a major factor in macrophage polarization from M1 to M2 type in vitro

LPS-stimulated macrophages cocultured with EVs showed reduced levels of TNF-α and increased IL-10 levels, compared to untreated and TSG-6-depleted EV-treated. Moreover, we

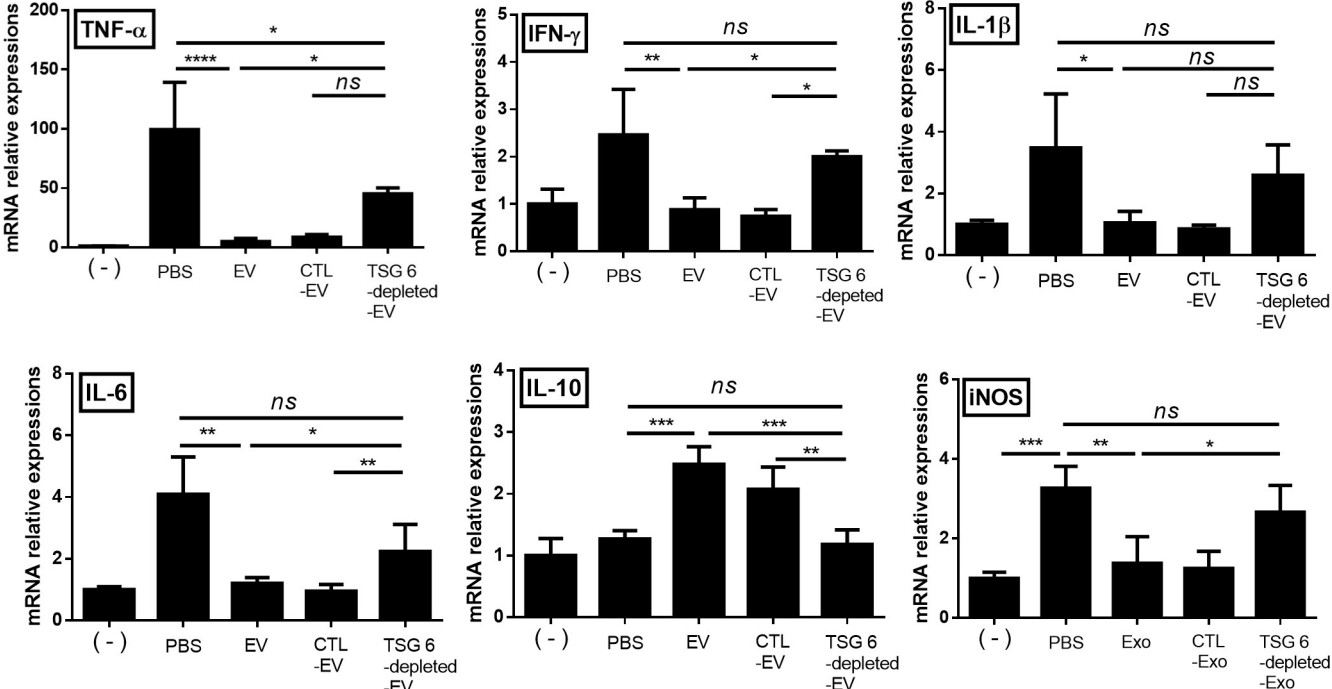

**Fig 3. EVs from cASCs inhibited inflammatory responses in the colon.** mRNA-expression levels of pro- and anti-inflammatory cytokines in the colon were determined by qRT-PCR. These data show that TSG-6 in EVs played a major role in regulating inflammatory cytokine levels in the colon. The results are shown as the mean ± standard deviation (n = 6–8 in each group, *P < 0.05, **P < 0.01, ***P < 0.001, as determined by one-way ANOVA).

found that CD206 and Arg mRNA expression levels increased in the EV group but decreased in the TSG-6 depleted EV group (Fig 5A). To determine the effect TSG-6 contained in EV on macrophage polarization, the extent of M1 and M2 were confirmed. Accordingly, the number of CD206+cells increased in the EV group. However, in the TSG-6-depleted EV group, the proportion of CD206+cells decreased (Figs 5B and S5). Contrary, the number of CD11c+cells decreased in the EV group compared to PBS group. However, in the TSG-6-depleted EV group, the proportion of CD11c+cells increased compared to EV group.

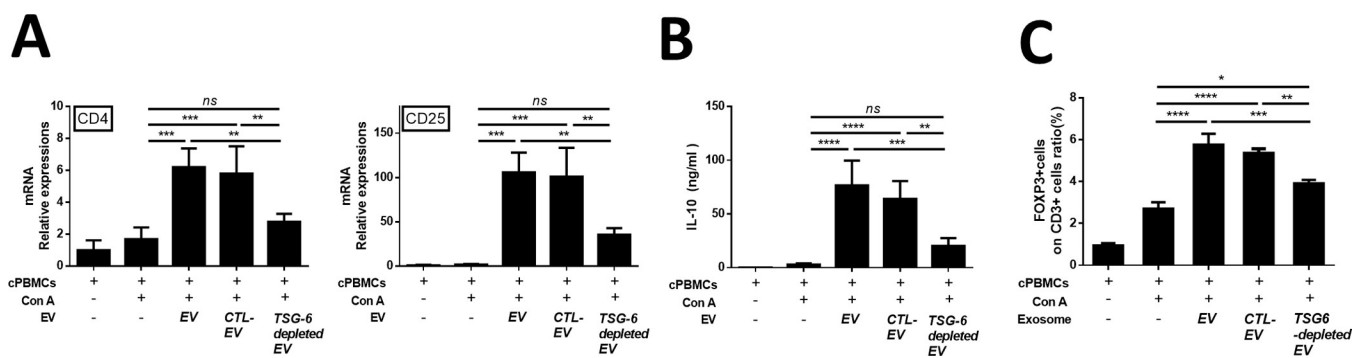

**Fig 4. cASC-EV TSG-6 increased Treg proliferation *in vitro*.** Con A-stimulated canine lymphocytes were cocultured for 48 h with cASC-EVs transfected with TSG-6 siRNA (si-TSG6) or scrambled siRNA (siCTL), or naïve EVs. (A) CD4 and CD25 mRNA-expression levels were measured, confirming that TSG-6 was associated with increased Treg production. (B) IL-10, which is known to be secreted from Tregs, was also measured in the supernatant medium, and the results confirmed that IL-10 production in lymphocytes was associated with TSG-6 (n = 6 in each group). (C) The Treg population was determined by measuring FOXP3 and CD3 double-positive cells by flow cytometry (n = 6 in each group). The results are presented as the mean ± standard deviation (*P < 0.05, **P < 0.01, ***P < 0.001, ****P < 0.0001, as determined by one-way ANOVA).

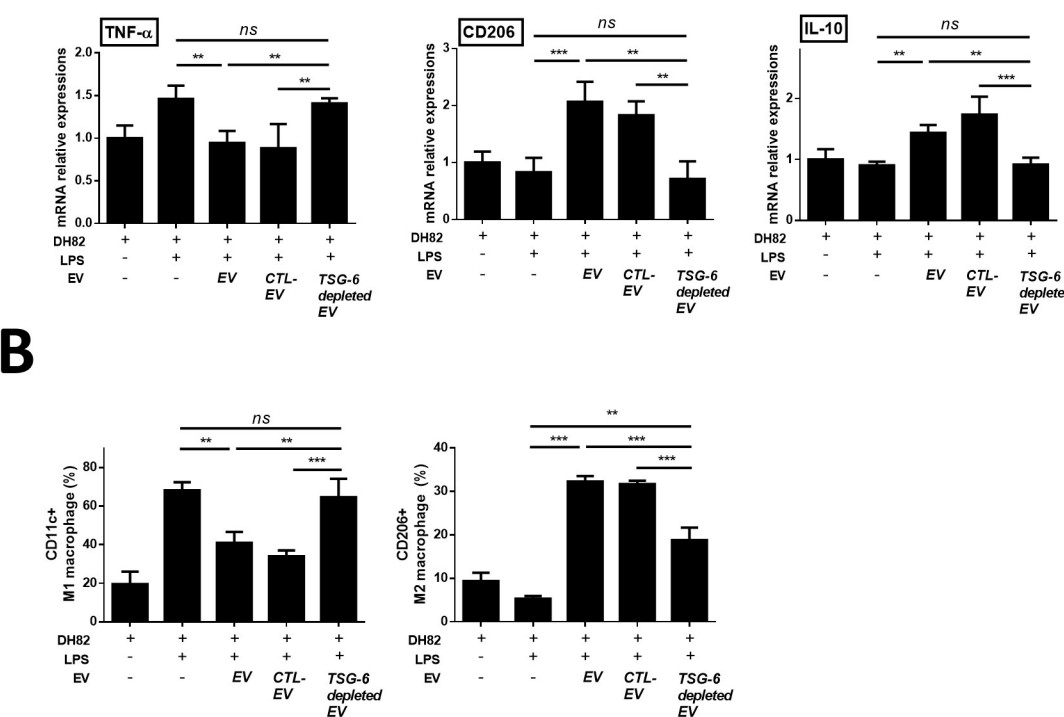

**Fig 5. cASC-EV TSG-6 induced macrophage polarization from M1 to M2 type in vitro.** LPS-stimulated canine macrophage (DH82) were cocultured for 48 h with cASC-EVs transfected with TSG-6 siRNA (si-TSG6) or scrambled siRNA (siCTL), or naïve EVs. (A) TNF-α, IL-10, CD206 and Arg mRNA-expression levels were measured (C) The M1 and M2 population were determined by measuring CD206 and CD11c positive cells by flow cytometry (n = 6 in each group). The results are presented as the mean ± standard deviation (*P < 0.05, **P < 0.01, ***P < 0.001, ****P < 0.0001, as determined by one-way ANOVA).

### TSG-6 in EV induced phenotypic enhancement of Tregs and M2 macrophage in inflamed colon

The FOXP3, CD4, CD25, CD206 and Arg mRNA levels were evaluated to examine whether stem cell EVs affected the activation of Tregs and polarization of macrophage in the inflamed colon. FOXP3 (6.8 fold), CD4 (3.0 fold), CD25 (4.0 fold), CD206 (3.0 fold) and Arg (2.7 fold) mRNA levels increased in the EV group compared to the PBS group. However, their levels significantly decreased in the TSG-6-depleted EV group compared to the EV and CTL-EV group (Fig 6A). To determine whether the increase in the number of Tregs and M2 was associated with TSG-6 in the EVs, quantitative analysis of FOXP3+ cells and CD206+ cells detected in colon tissue sections by immunofluorescence showed that the percentage of FOXP+ cells (4.76 fold) and (7.61 fold) increased significantly in the cASC-EV group compared to that in the PBS group. However, the enhancement of the number of FOXP3+ cells (0.54 fold) and CD206+ cells (0.35 fold) in the EVs of the colon tissue decreased when TSG-6 was inhibited (Fig 6B).

## Discussion

Our data suggests the following important points: (1) TSG-6 in the stem cell derived EVs is a key factor in immune regulation and relieving inflammation in the DSS-induced mouse

**A**

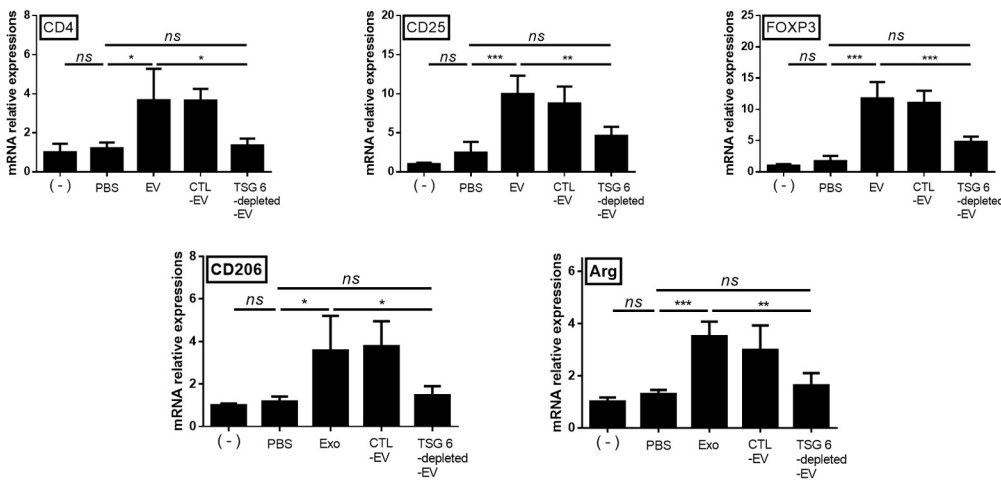

**B**

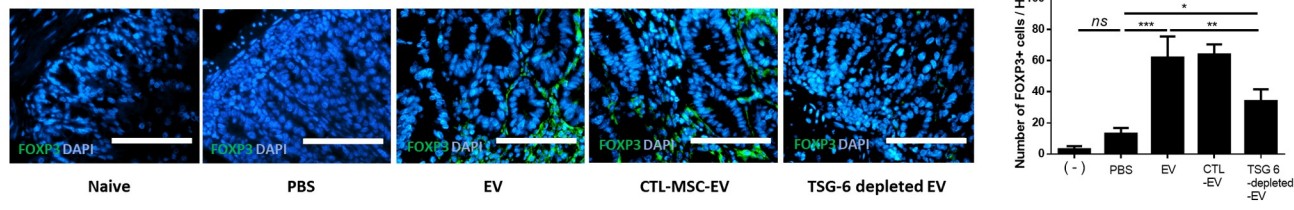

**C**

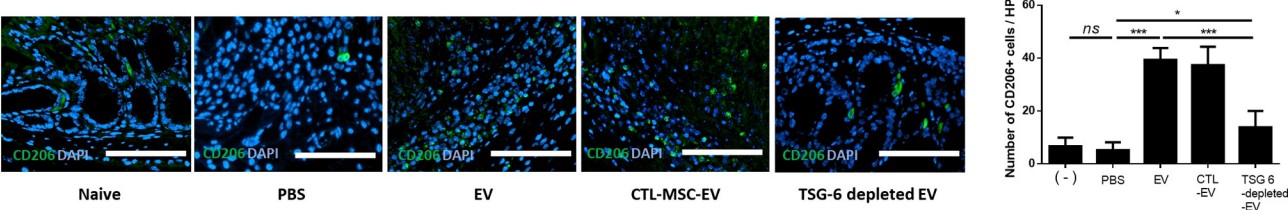

**Fig 6. TSG-6 increases regulatory T cells in the inflamed colon.** TSG-6 in EVs increased the proportion of Tregs in the inflamed colon. (A) Relative gene-expression levels of CD4, CD25, FOXP3, CD206 and Arg in the inflamed colon. (B) FOXP3+ (green) cells (C) CD206+ (gren)cells were detected in colon tissue sections by immunofluorescence. The data shown demonstrated that TSG-6 in EVs played a major role in increasing the number of Tregs and M2 in the colon. Scale bar, 50 μm. The results are shown as the mean ± standard deviation (n = 6–8 in each group, *P < 0.05, ****P < 0.0001, as determined by one-way ANOVA).

model of colitis. (2) TSG-6 in EVs alleviates inflammation by enhancing colonic Tregs and polarizing colonic macrophage from M1 and M2 in an IBD mouse model.

Previous studies have shown that stem cells affect recipient cells in a paracrine manner, considering that EVs largely account for the paracrine effect of stem cells [3, 4, 30]. In our, preliminary study, the immune regulatory capacity of stem cells was significantly decreased when

GW4869 (a noncompetitive neutral sphingomyelinase (N-SMase) inhibitor; exosome inhibitor) was applied to cASCs (S3 Fig). Moreover, EVs and stem cells have similar immunomodulatory effects (S4 Fig).

In addition, with this tendency, various studies have been conducted since the introduction of EV as a therapeutic agent [3, 31]. However, the exact mechanism by which they relieve colitis has not been revealed. Therefore, it is noteworthy that this study demonstrated that TSG-6 in EVs is a major factor in relieving colitis symptoms.

Previous studies have shown that immune cells play a role in controlling inflammation in the colon [32]. Among these immune cells, Tregs have been described as having important roles in regulating the pathogenesis of IBD [33, 34] and the balance between Tregs and other T cells in the intestinal tract is known to influence IBD pathogenesis [35]. To a large extent, FOXP3 amplifies and stabilizes the molecular features of Treg precursor cells, which is beneficial for their function and maintenance, and attenuates features that are deleterious to Treg functions [36]. FOXP3-expressing Tregs, which belong to a suppressive subset of CD4[+] T cells, can regulate infection, tumor development, allergy, and autoimmunity [37]. It was reported that FOXP3[+] Tregs are lower in patients with IBD progression than in healthy controls. It was also reported that an increase in the number of Tregs after treating IBD patients correlated with relief of IBD symptoms [35]. In other words, increasing the number of Tregs may serve as a method for treating IBD.

Colon macrophages are essential for maintaining mucosal homeostasis for the ongoing need for epithelial regeneration but are also an important component of protective immunity and are involved in the pathology of IBD [38]. M1 preferentially metabolize arginine to nitric oxide via inducible nitric oxide synthase (iNOS; NOS2), while M2 preferentially metabolize arginine to ornithine via arginase -1. Therefore, it is known that M1 are involved in pre-inflammatory and M2 are involved in anti-inflammatory [39]. The importance of macrophages in maintaining immune homeostasis has shown that IL-10 secreted from M2 acts on Treg to maintain Foxp3 expression [40]. These Foxp3-expressed Tregs relieved inflammation by suppressing the activity of Th1 and Th17 cells [41]. Therefore, increasing M2 in the colon is noted as a way to alleviate colitis [21]. Accumulated evidence shows that infusion of EVs into colitis models relieves inflammation [42], but the relationship between colonic immune cells and EVs is not clear. Therefore, our study is valuable in that it clarifies the relationship between TSG-6 in EV and colonic immune cells such as Tregs and macrophages.

Although proteins other than TSG-6, such as TGF-beta, IDO and PGE2 may also be contributing to the protective effect of EVs in relieving inflammation [43]. Previous studies have reported that TGF-β plays a role in inhibiting activated immunity by inducing FoxP3+ regulatory t cells in an inflammatory environment and has been shown to play an important role in relieving inflammation in colitis models [44]. Zhang et al. reported that TGF-beta in bone marrow derived stem cells plays a major role in polarizing macrophage from M1 to M2 [45]. IDO expression and activity is an important mediator of intestinal homeostasis both in health and disease [46]. In addition, IDO appears to be the most promising candidate, which plays an important role in the immunomodulatory effects of stem cells by inhibiting T cell activation and enhancing Tregs [47]. Also, IDO has been shown to play a major role in suppressing immunity by polarizing macrophage from M1 to M2.[48]. Furthermore, IDO as an anti-inflammatory agent has been reported to reduce inflammation in colitis models [49]. In addition, in our previous study, we confirmed the efficacy of adipose-derived stem cells in murine-derived macrophage cell lines in inflammatory environments and demonstrated that PGE2 secreted from stem cells is a key factor in polarizing macrophage [50, 51]. Moreover, we previous showed that PGE2 secreted feline ASC is a key factor for enhancing regulatory T cell in inflamed colon [52]

Although further research on the correlation between these immunomodulatory factors of EV and immune cell regulation is needed, results of the current study confirm that TSG-6-depleted EVs significantly reduce the immunoregulatory ability, which clearly indicates that TSG-6 is a major factor in immune regulation and anti-inflammatory action. Furthermore, the finding that TSG-6 in EVs plays an important role in immune regulation will serve as evidence to support increasing the level of TSG-6 in EVs as a strategy to develop EVs with enhanced immunomodulating properties.

Although EV-specific studies have not been conducted, other studies have shown that pretreatment of stem cells with TNF-alpha (TNF-α) resulted in an increase in mRNA levels of TSG-6 in stem cells as well as increased levels of TSG-6 protein in the culture medium [21, 53]. Further studies on pre-treated stem cell-derived EVs are needed.

Nuclear transcription factor kappaB (NF-κB) is a central mediator of pro-inflammatory gene induction and function in immune cells and has a significant effect on mucosal inflammatory process [54]. Moreover, in IBD patients, its activation is markedly induced. Therefore, the NF-κB pathway is considered to be an attractive target of therapeutic intervention in IBD [55]. In our previous study, TSG-6 from stem cells significantly suppressed nuclear factor kappa B (NF-κB) activity and alleviated inflammation and reduced apoptosis in acute pancreatitis model [20]. However, the relationship between EV and NF-kB has not been studied in this study, and it is necessary to confirm whether EV's TSG-6 lowers NF-kB in inflammatory colon.

This study is also an important basis for future transitional studies. Like human IBD, canine idiopathic IBD is a commonly observed chronic IBD that occurs spontaneously with similar multifocal etiology due to the interactions between abnormal host immune responses, and genetic and environmental factors. Histological evaluation of intestinal biopsies reveals extensive or multifocal inflammatory cell infiltration (most commonly lymphoid evolutive, eosinophilic, and neutrophilic), with simultaneous changes in mucosal structures (e.g. villous atrophy and fusion). In severe cases, intestinal protein loss, similar to in human disease, can be observed [1, 56, 57]. This study has been carried out with EVs derived from canine cells and is of great value in facilitating subsequent experiments in dogs. Therefore, evaluating the efficacy of TSG-6 in EVs conducted in this study is valuable for applications in veterinary medicine, particularly for intractable immune-mediated diseases such as IBD, however, much of these finds may also be applicable to human IBD in the future.

## Conclusion

We demonstrated that TSG-6 in EVs secreted from cASCs ameliorated DSS-induced colitis in mice by enhancing the Treg population and polarizing macrophage from M1 to M2 in the inflamed colon. Our findings provide an insight to improve the current understanding of the role that EVs have in immunoregulation and serve as a foundation for applying EVs as a therapeutic agent in IBD. Also, this study is the basis of a strategy for developing EVs with improved immunomodulatory properties by increasing TSG-6 levels in EVs.

## Supporting information

**S1 Fig. Identification of cASCs and si TSG-6 cASC.** (A) Immunophenotypes of the cultured cASCs were examined by flow cytometry. The vast majority of cells were positive for CD90, CD44, CD29, and CD73, but a few cells expressed CD34 and CD45. (B) The naïve and si TSG-6 cASCs were maintained in specific differentiation media for 3 weeks, and the differentiated cells were stained by Oil red O to identify adipocytes, Alizarin Red S for osteocytes and Alcian Blue for chondrocytes. Scale bars, 200 μm. (C) Cell-viability assays of naïve and siTSG6-cASCs.

si TSG-6 transfection was not cytotoxic when applied to stem cells. (n = 6 in each group) (D) Morphology of EVs from siTSG6-cASCs, as studied by transmission electron microscopy. EV was identified as a circular particle with a diameter of less than 100 nm. (E) EV production by naïve and siTSG6-cASCs. The production of exosome does not differ between naive and siTSG-6 groups. (n = 6 in each group) The results are shown as the mean ± standard deviation (ns, not significant, were analyzed using Student's t-tests)
(TIF)

**S2 Fig. Production of TSG-6 depleted EV.** (A) TSG-6 mRNA-expression levels in naïve cASCs, cASCs transfected with a scrambled siRNA (CTL-cASC), or cASCs transfected with TSG-6 (siTSG-6-cASC) was determined by agarose gel electrophoresis and RT-qPCR. (Lane 1 and 2: Naïve, Lane 3 and 4: CTL-cASC, Lane 5 and 6: si TSG-6 cASC in gel PCR) (B) TSG-6 protein-expression levels in naïve cASC-EVs, EVs from cASCs transfected with a scrambled siRNA (CTL-EV), or EVs from cASCs transfected with TSG-6 (TSG-6 depleted-EV) were determined by western blot analysis. The results are presented as the mean ± standard deviation. (n = 6 in each group) (*ns* = Not Statistically Significant *P < 0.05, **P < 0.01, ***P < 0.001 by one-way ANOVA analysis).
(TIF)

**S3 Fig. Immunological biomarkers observed upon co-culturing total lymphocytes with cASCs.** (A) Treatment with 0.005% DMSO, 10 μM, 20 μM GW4869, or 1% DMSO showed no cytotoxic effects on cASCs, as shown by similar viability rates following all treatments, compared to the non-treated group (n = 6 in each group) (B) Pre-treatment with GW4869(10 μM, for 12h) significantly reduced production of EV proteins by cASCs. EV production was reduced by more than 70% in the GW4869-treated group (n = 6 in each group ) (* p< 0.05, were analyzed using Student's t-tests)(C) The mRNA levels of TNA-α, IL-1β, IL-6, IFN-γ, and IL-10 were detected by qRT-PCR. Con A-treated lymphocytes showed significantly increased levels of pro-inflammatory cytokines, such as TNF-α, IFN-γ, IL-1β, and IL-6, compared to the untreated group. cASCs depressed activated lymphocyte. however, pre-treatment with GW4869 significantly reduced the modulatory effects of cASCs. (n = 6 in each group)The results are presented as the mean ± standard deviation (**P < 0.01, ***P < 0.001, ****P < 0.0001 as determined by one-way ANOVA).
(TIF)

**S4 Fig. Immunomodulatory effects of cASC-EVs.** (A) Changes in the expression levels of mRNAs encoding several canine lymphocyte-derived cytokines including TNF-α, IL-1β, IFN-γ, IL-6, and IL-10 in the presence of cASC-Evs (100ug/well). After Con A-stimulated lymphocytes were cocultured with EV (100 ug), the levels of activated pro-inflammatory cytokines (TNF-α, IFN-γ, IL-1β, and IL-6) decreased significantly. Production of the anti-inflammatory cytokine IL-10 significantly increased, compared to that in the untreated group. (B) Changes in the expression levels of mRNAs encoding several canine macrohage-derived cytokines including TNF-α, IL-6, iNOS and IL-10 in the presence of cASC-Evs (100ug/well). After LPS-stimulated DH82 were cocultured with EV (100 ug), the levels of activated pro-inflammatory cytokines (TNF-α, IL-6 and iNOS) decreased significantly. Production of the anti-inflammatory cytokine IL-10 significantly increased, compared to that in the untreated group. The data show that EVs exerted immunosuppressive effects as much as stem cells. The results are presented as the mean ± standard deviation (n = 6 in each group), **P < 0.01, ***P < 0.001, ****P < 0.0001, as determined by one-way ANOVA).
(TIF)

**S5 Fig. TSG-6 in EV enhance regulatory T cells and regulate the M1/M2 balance in vitro.** TSG-6 in EV plays an important role in the increase of regulatory t cells and macrophage polarization. (A) Tregs (FOXP3+CD3+ cells) level in canine lymphocytes (B) M1 (CD11+cells) and M2 macrophages (CD206+ cells) level in canine macrophage cell line (DH82). FACS plots (right panel) show representative examples and bar graphs (left panel) represent mean values +SD (*ns* = Not Statistically Significant *P < 0.05, **P < 0.01, ***P < 0.001 by one-way ANOVA analysis)
(TIF)

**S1 Raw Images. protein marker of cASC derived EV were analysis by western blot.** The original underlying images of CD63, CD9, Lamin A and β-actin in Fig 1D.
(TIF)

**S1 File. Isolation, Culture and Characterization of cASCs.**
(DOCX)

## Author Contributions

**Conceptualization:** Ju-Hyun An, Qiang Li, Woo-Jin Song, Hwa-Young Youn.

**Data curation:** Ju-Hyun An, Min-Ok Ryu, A-Ryung Nam.

**Formal analysis:** Ju-Hyun An, Min-Ok Ryu, A-Ryung Nam.

**Funding acquisition:** Hwa-Young Youn.

**Investigation:** Ju-Hyun An, Dong-Ha Bhang, Yun-Chan Jung.

**Resources:** Ju-Hyun An, Woo-Jin Song, Hwa-Young Youn.

**Supervision:** Woo-Jin Song, Hwa-Young Youn.

**Validation:** Ju-Hyun An.

**Visualization:** Ju-Hyun An.

**Writing – original draft:** Ju-Hyun An.

**Writing – review & editing:** Woo-Jin Song, Hwa-Young Youn.

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
