## [Decision Letter · Decision Letter 0]

4 Nov 2019

PONE-D-19-20533

TSG-6 in extracellular vesicles from canine mesenchymal stem/stromal is a major factor in relieving DSS-induced colitis

PLOS ONE

Dear Dr. Youn,

Thank you for submitting your manuscript to PLOS ONE. Your submission has been thoroughly peer-reviewed by two independent referees. After careful consideration, we feel that it has merit but does not fully meet PLOS ONE’s publication criteria as it currently stands. Therefore, we invite you to submit a revised version of the manuscript that addresses the points raised during the review process.

Please respond ASAP to the queries raised by reviewers and given hereunder. Besides, the following shortcomings should be addressed when revising the manuscript:

1- One should specify the animal sample size for each group enrolled in the study (Line 6; "the mice were randomly divided into 6 groups with 6-8 mice per group"- misleading and not accepted).

2- In the statistical analysis section; Line 14, the criterion for least significance (p < 0.05) should be narrated. The individual t-tests, in contrast to multiple comparison post-hoc tests, can estimate the variances only from 2 of the 6 groups, which is less precise and less reliable. Since the variance is estimated from the whole set of data (6 groups) as pooled estimate, such estimate is much more robust and precise than that estimated from just a part of the whole set of data. So, authors ought to use less conservative and more accurate post-hoc test such as Tukey or Bonferroni. As regards the histological scoring system, did the authors perform any non-parametric multiple comparison tests in similar analogy to ANOVA? if so- which test did they perform?

3- In results, one should not discuss his findings just narrate them. Hence, the following statements should be omitted and transferred to the discussion section:

- Lines 14-16/page 15; "Collectively, these results indicate that EV 14 administration alleviated DSS-induced colitis and indicate that TSG-6 in EV played a role in relieving colitis".

- Last paragraph in page 15; "These results indicate that EV markedly attenuated the inflammatory state in mice with induced colitis and suggest that TSG-6 in EV played an important role in relieving inflammatory conditions".

3- Lines 4-9/page 16; there is no room in results for such verbosity "TSG-6 was a major factor in increasing Tregs in Con A stimulated lymphocytes In vitro.The expression levels of CD4, and CD25 mRNA in lymphocytes derived canine PBMCs were assessed to determine whether TSG-6 in EVs affected Treg activation. Con A, a mannose/glucose-binding lectin, is a well-known T cell mitogen that can activate the immune system, recruit lymphocytes, and elicit cytokine production. Therefore, lymphocytes were stimulated with Con A to confirm the immune cell-control function of EVs". Authors either should omit it or incorporate in the introduction section.

4- Lines 1-2/page 17;  the sentence reads "FOXP3, CD4, and CD25 mRNA levels increased in the EV group. However, their levels decreased in the TSG-6-depleted EV group". Authors should specify the percentage changes whether steered up or down. 

5- Lines 5-7/page 17; the same holds true for the sentence  "the percentage of these cells increased significantly in the cAT-MSC-EV group compared to that in the PBS group. However, the enhancement of the number of FOXP3+ cells in the EVs of the colon tissue decreased when TSG-6 was inhibited". One should exactly clarify the percentage increase or decrease.

We would appreciate receiving your revised manuscript by 30 days. To enhance the reproducibility of your results, we recommend that if applicable you deposit your laboratory protocols in protocols.io, where a protocol can be assigned its own identifier (DOI) such that it can be cited independently in the future. For instructions see: http://journals.plos.org/plosone/s/submission-guidelines#loc-laboratory-protocols

We look forward to receiving your revised manuscript.

Kind regards,

Hossam MM Arafa

Academic Editor

PLOS ONE

Journal Requirements:

3. In your Methods section, please provide additional details regarding the mice used in your study and ensure you have described the source. For more information regarding PLOS' policy on materials sharing and reporting, see https://journals.plos.org/plosone/s/materials-and-software-sharing#loc-sharing-materials.

6. Thank you for stating the following in the Financial Disclosure section:

H.Y. YOUN

The Research Institute for Veterinary Science, Seoul National University and Basic Science Research Program of the National Research Foundation of Korea

http://vetsci.snu.ac.kr/

These funds contributed to the collection, analysis, and interpretation of data generated in this study.

We note that one or more of the authors are employed by a commercial company: Chaon Corporation

7. Your ethics statement must appear in the Methods section of your manuscript. If your ethics statement is written in any section besides the Methods, please move it to the Methods section and delete it from any other section. Please also ensure that your ethics statement is included in your manuscript, as the ethics section of your online submission will not be published alongside your manuscript.

Reviewers' comments:

Reviewer's Responses to Questions

**Comments to the Author**

1. Is the manuscript technically sound, and do the data support the conclusions?

Reviewer #1: Yes

Reviewer #2: Partly

2. Has the statistical analysis been performed appropriately and rigorously? 

Reviewer #1: Yes

Reviewer #2: Yes

3. Have the authors made all data underlying the findings in their manuscript fully available?

Reviewer #1: Yes

Reviewer #2: Yes

4. Is the manuscript presented in an intelligible fashion and written in standard English?

Reviewer #1: Yes

Reviewer #2: Yes

5. Review Comments to the Author

Reviewer #1: I would like to thank the authors for their great efforts in conducting the study in this well organised manner.

Overall, the manuscript emphasises on using EVs as a next generation therapeutic agent as an alternative to stem cells for immune-mediated diseases owing to their similar immunomodulatory effect which partly dependent on TSG6. This is beside the notable anti inflammatory effect exerted by EVs that was also attiributed to TSG6. The results were promising and provide an important basis for future transitional studies.

However, i have some minor comments:

1-What is the cause of investigating the effect of TSC6 in EVs on Tregs only without investigating the effect on other immune cells like M2 macrophages switch? A previous study in 2017 states that hAT-MSC-produced TSG-6 can ameliorate IBD by inducing M2 macrophage switch in mice.

2-What is the cause of selecting TSG6 although proteins other than TSG-6, such as TGF-beta, IDO, PGE2, and NO, may also be contributing to the protective effect of EVs in relieving inflammation? A detailed justification may be required.

3-Increasing the level of TSG-6 in EVs as a strategy to develop EVs with enhanced immunomodulating properties is thought to be a very promising idea that needs to be addressed as a major recommendation or even to be added in the conclusion section.

Some typographical errors:

The abbreviation TEM (page 14, line 17)is not included in the abbreviations list.

Page 13, line 9; as previously described not as previous.

Page 24, line 12; immune mediated diseases not disease.

Thank you

Best regards

Reviewer #2: you write in the experimental design that is 6 groups but in some figure 5 group only

CTl , si-tsg-6 are not present in the abbreviation list and not mentioned in experimental design

what is the difference between CTL-MSc-EV and TSG 6 depleted EV. what is the significance of using these two groups

experimental design needs to clarify (page no 12)

there is very high S.D in the results such as figure 4, 5 and 6.

why you didn't use DSS 5%

why you didn't analyze the stool EV, colon length

also why you didn't measure NF-KBp65, INOs, caspase 3

is a relation between TSG 6 and TSG 14 in colitis

in reference No 9 (the paper on renal not on colon ) you need to check the paper

https://www.ncbi.nlm.nih.gov/pmc/articles/PMC3303802/

6. PLOS authors have the option to publish the peer review history of their article (what does this mean?). If published, this will include your full peer review and any attached files.

Reviewer #1: No

Reviewer #2: No

---

## [Author Response · Author response to Decision Letter 0]

25 Nov 2019

November 15, 2019 

Hossam MM Arafa

Academic Editor

PLOSONE

Dear Prof. Joerg Heber

We are very pleased to have been given the opportunity to revise our manuscript entitled “TSG-6 in extracellular vesicles from canine mesenchymal stem/stromal is a major factor in relieving DSS-induced colitis” for PLOSONE. We want to extend our appreciation to you and the reviewers for taking the time and effort necessary to provide such insightful guidance. We have carefully considered comments offered by the reviewers. Herein, we explain how we revised the paper based on those comments and recommendations. The manuscript has certainly benefited from these revision suggestions. We look forward to working further with you and the reviewers to move this manuscript closer to publication.

COMMENT 1- One should specify the animal sample size for each group enrolled in the study (Line 6; "the mice were randomly divided into 6 groups with 6-8 mice per group"- misleading and not accepted).

RESPONSE: we revised this in abstract section. 

“ DSS-induced colitis mice (C57BL/6, male, Naïve n= 6, Sham n =8, PBS n=8, EV n=8, CTL-EV n=8, TSG-6 depleted EV n=8) were intraperitoneally administered EVs (100 ug/mice) on day 1, 3, and 5; colon tissues were collected on day 10 for histopathological, RT-qPCR, western blot, and immunofluorescence analyses”

COMMENT 2- In the statistical analysis section; Line 14, the criterion for least significance (p < 0.05) should be narrated. The individual t-tests, in contrast to multiple comparison post-hoc tests, can estimate the variances only from 2 of the 6 groups, which is less precise and less reliable. Since the variance is estimated from the whole set of data (6 groups) as pooled estimate, such estimate is much more robust and precise than that estimated from just a part of the whole set of data. So, authors ought to use less conservative and more accurate post-hoc test such as Tukey or Bonferroni. As regards the histological scoring system, did the authors perform any non-parametric multiple comparison tests in similar analogy to ANOVA? if so- which test did they perform?

RESPONSE: we revised this in the statistical analysis section

“GraphPad prism (version 6.01) software (GraphPad, Inc., La Jolla, CA, USA) was used for statistical analysis. the differences between two groups were analyzed using Student’s t-tests and differences between more than two groups were analyzed using one-way analysis of variance (ANOVA) followed by Bonferroni multiple comparison test. The results are presented as the mean value ± standard deviation (SD). Differences with a value of P < 0.05 were considered as statistically significant.”

And the histological scoring section, we analyzed the group ANOVA except the naive and sham groups. The reason why the PBS group was compared with the naive group was to compare the well-induced colitis model with DSS and to see if EV has significantly improved DAI. I hope you agree. 

COMMENT 3- In results, one should not discuss his findings just narrate them. Hence, the following statements should be omitted and transferred to the discussion section:

Lines 14-16/page 15; "Collectively, these results indicate that EV 14 administration alleviated DSS-induced colitis and indicate that TSG-6 in EV played a role in relieving colitis".

Last paragraph in page 15; "These results indicate that EV markedly attenuated the inflammatory state in mice with induced colitis and suggest that TSG-6 in EV played an important role in relieving inflammatory conditions".

RESPONSE: Thank you for the good comment. This paragraph has moved to the discussion section.

COMMENT 4- Lines 4-9/page 16; there is no room in results for such verbosity "TSG-6 was a major factor in increasing Tregs in Con A stimulated lymphocytes In vitro.The expression levels of CD4, and CD25 mRNA in lymphocytes derived canine PBMCs were assessed to determine whether TSG-6 in EVs affected Treg activation. Con A, a mannose/glucose-binding lectin, is a well-known T cell mitogen that can activate the immune system, recruit lymphocytes, and elicit cytokine production. Therefore, lymphocytes were stimulated with Con A to confirm the immune cell-control function of EVs". Authors either should omit it or incorporate in the introduction section.

RESPONSE: Thank you for the good comment. This paragraph has deleted in this section 

COMMENT 5- Lines 1-2/page 17; the sentence reads "FOXP3, CD4, and CD25 mRNA levels increased in the EV group. However, their levels decreased in the TSG-6-depleted EV group". Authors should specify the percentage changes whether steered up or down. 

RESPONSE: Thank you for the good comment. We revised this in result section. Revised paragraph is described below. 

“The FOXP3, CD4, CD25, CD206 and Arg mRNA levels were evaluated to examine whether stem cell EVs affected the activation of Tregs in the inflamed colon. FOXP3 (6.8 fold), CD4(3.0 fold), CD25 (4.0 fold), CD206 (3.0 fold) and Arg (2.7 fold) mRNA levels increased in the EV group compared to the PBS group. However, their levels significantly decreased in the TSG-6-depleted EV group compared to the EV and CTL-EV group (Figure 6A).”

COMMENT 6- Lines 5-7/page 17; the same holds true for the sentence "the percentage of these cells increased significantly in the cAT-MSC-EV group compared to that in the PBS group. However, the enhancement of the number of FOXP3+ cells in the EVs of the colon tissue decreased when TSG-6 was inhibited". One should exactly clarify the percentage increase or decrease.

RESPONSE: Thank you for the good comment. We revised this in result section. Revised paragraph is described below.

“To determine whether the increase in the number of Tregs was associated with TSG-6 in the EVs, quantitative analysis of FOXP3+ cells detected in colon tissue sections by immunofluorescence showed that the percentage of these cells (4.76 fold) increased significantly in the cAT-MSC-EV group compared to that in the PBS group. However, the enhancement of the number of FOXP3+ cells in the EVs of the colon tissue (0.54 fold) decreased when TSG-6 was inhibited”

 

Ⅰ Journal Requirements:

COMMENT 1. When submitting your revision, we need you to address these additional requirements. Please ensure that your manuscript meets PLOS ONE's style requirements, including those for file naming. The PLOS ONE style templates can be found at

RESPONSE: we revised this in revised manuscript. 

COMMENT2. PLOS ONE now requires that authors provide the original uncropped and unadjusted images underlying all blot or gel results reported in a submission’s figures or Supporting Information files. This policy and the journal’s other requirements for blot/gel reporting and figure preparation are described in detail at https://journals.plos.org/plosone/s/figures#loc-blot-and-gel-reporting-requirements and https://journals.plos.org/plosone/s/figures#loc-preparing-figures-from-image-files. When you submit your revised manuscript, please ensure that your figures adhere fully to these guidelines and provide the original underlying images for all blot or gel data reported in your submission. See the following link for instructions on providing the original image data: https://journals.plos.org/plosone/s/figures#loc-original-images-for-blots-and-gels.

RESPONSE: we revised this in revised manuscript. These raw files were named S1_raw images (contained with figure 1 contents) and uploaded as a Supporting Information file. 

COMMENT3. In your Methods section, please provide additional details regarding the mice used in your study and ensure you have described the source. For more information regarding PLOS' policy on materials sharing and reporting, see https://journals.plos.org/plosone/s/materials-and-software-sharing#loc-sharing-materials.

RESPONSE: we revised this in revised manuscript as follow in M&M section.

“To determine the therapeutic effect of ASC-derived EVs on colitis, we induced mouse colitis with DSS (36–50 kDa; MP Biomedical, Solon, OH, USA). Male C57BL/6 mice (6 to 8-week-old and weighing 18 to 20 g) were purchased from Nara Bio (Gyeonggi, Korea) and acclimatized for 7 days with a 12 h light/dark cycle at 22 ℃ and 60 % humidity before performing the experiments. For environmental enrichment, 3 to 4 mice were raised in polycarbonate cages (324 × 221.5 × 130 mm) containing clean bedding (shavings; Nara Biotech), cardboard boxes, and tunnels. Mice were fed a standard laboratory rodent diet (Central Lab Animal Inc., Seoul, Korea) and water ad libitum.”

COMMENT4. Please include captions for your Supporting Information files at the end of your manuscript, and update any in-text citations to match accordingly. Please see our Supporting Information guidelines for more information: http://journals.plos.org/plosone/s/supporting-information.

RESPONSE: we revised this in revised manuscript. 

COMMENT5. We note that you have included the phrase “data not shown” in your manuscript. Unfortunately, this does not meet our data sharing requirements. PLOS does not permit references to inaccessible data. We require that authors provide all relevant data within the paper, Supporting Information files, or in an acceptable, public repository. Please add a citation to support this phrase or upload the data that corresponds with these findings to a stable repository (such as Figshare or Dryad) and provide and URLs, DOIs, or accession numbers that may be used to access these data. Or, if the data are not a core part of the research being presented in your study, we ask that you remove the phrase that refers to these data.

RESPONSE: Thank you for the good point. This phrase has been removed.

COMMENT 6. Thank you for stating the following in the Financial Disclosure section:

H.Y. YOUN

The Research Institute for Veterinary Science, Seoul National University and Basic Science Research Program of the National Research Foundation of Korea

http://vetsci.snu.ac.kr/

These funds contributed to the collection, analysis, and interpretation of data generated in this study.

We note that one or more of the authors are employed by a commercial company: Chaon Corporation

RESPONSE: Thank you for your attention. In this study, funding was received only by HYY, and the rest is not applicable to funding. We described as follow

Conflicts of interest

The authors declare that no conflicts of interest exist regarding the publication of this article.

Availability of data and material

The datasets used and/or analyzed during the current study are available from the corresponding author on reasonable request.

Financial Disclosure 

H.Y. YOUN

The Research Institute for Veterinary Science, Seoul National University and Basic Science Research Program of the National Research Foundation of Korea. These funds contributed to the collection, analysis, and interpretation of data generated in this study.

Acknowledgements

Not applicable

Authors’ contributions

JHA conceived and designed the study; collected, analyzed, and interpreted the data; and helped in writing the manuscript. QL participated in the conception and design of the study. MOR and ARN collected the data. DHB and YCJ provided administrative support, supported study materials and collected data. WJS and HYY contributed to the conception and design of the study, data analysis and interpretation, and granted final approval of the manuscript. All authors have read and approved the final manuscript.

And we described as follow in cover letter

“And the authors declare that no conflicts of interest exist regarding the publication of this article. This study is partially supported by the Research Institute for Veterinary Science, Seoul National University and Basic Science Research Program of the National Research Foundation of Korea.”

COMMENT 7. Your ethics statement must appear in the Methods section of your manuscript. If your ethics statement is written in any section besides the Methods, please move it to the Methods section and delete it from any other section. Please also ensure that your ethics statement is included in your manuscript, as the ethics section of your online submission will not be published alongside your manuscript.

RESPONSE: we revised this in revised manuscript. 

 

Ⅱ. Review Comments to the Author

Reviewer #1: 

COMMENT 1-What is the cause of investigating the effect of TSC6 in EVs on Tregs only without investigating the effect on other immune cells like M2 macrophages switch? A previous study in 2017 states that hAT-MSC-produced TSG-6 can ameliorate IBD by inducing M2 macrophage switch in mice.

RESPONSE: Thank you for your good feedback. During the under-review period, additional experiments were conducted, and it was confirmed that TSG-6 in EV not only increased regulatory T cells but also polarized macrophage to M2 type. In this regard, additional primers were added, additional in vitro experiments were performed using canine macrophage cell line, and macrophage related factors were additionally identified in colitis-induced mice. We described this as follow and in revised manuscript. We hope our approach acceptable.

1. in M&M section 

Co-culture experiments

DH82 cells, a canine macrophage-like cell line, were purchased from the Korean Cell Line Bank (Seoul, Korea). DH82 cells were seeded in 6-well plates (1×106 cells/well), then incubated for 24 h. After adherence to the plates was confirmed, the DH82 cells were treated with LPS (200 ng/mL; Sigma-Aldrich) or control for 24 h. Similarly, canine lymphocytes were seeded in 6-well plates (1×106 cells/well) and exposed to Con A (5 μg/mL) or control for 24 h. Next, the medium was removed and replaced with media containing EV (100 μg/well) derived from naïve, si RNA and si TSG-6 cAT-MSCs. Next, the cells were incubated for 48 h and then harvested for RNA extraction and flow cytometry analysis.

Table 1

ciNOS GAGATCAATGTCGCTGTACTCC TGATGGTCACATTTTGCTTCTG

cCD206 GGAAATATGTAAACAGGAATGATGC TCCATCCAAATAAACTTTTTATCCA

miNOS AAAGGAAATAGAAACAACAGGAACC GCATAAAGTATGTGTCTGCAGATGT

mCD206 AACGGAATGATTGTGTAGTTCTAGC TACAGGATCAATAATTTTTGGCATT

mArg CAGAAGAATGGAAGAGTCAG CAGATATGCAGGGAGTCACC

Flow cytometry analysis

To evaluate Treg polarization, PBMC-derived lymphocytes cocultured with cAT-MSC-EVs were harvested. Obtained cells (1 × 106) were suspended in 100μL DPBS and 1 μL of each primary antibody against the following proteins: FOXP3-PE (eBioscience, San Diego, CA, USA; 1:100), CD3-FITC (MCA1774F; Bio-Rad, San Diego, CA, USA; 1:100), CD206-FITC (eBioscience, San Diego, CA, USA; 1:100) and CD11c-PE (eBioscience, San Diego, CA, USA; 1:100). After incubation for 1 h at 23 ± 2 ℃, the cells were washed with DPBS. Unstained cells were used as controls for autofluorescence. Cell fluorescence was analyzed with a flow cytometer (FACS Aria Ⅱ; BD bioscience). The results were analyzed using FlowJo 7.6.5 software (Tree Star, Inc., Ashland, OR, USA).

2. in Results section 

“TSG-6 in EV is a major factor in macrophage polarization from M1 to M2 type in vitro 

LPS-stimulated macrophages cocultured with EVs showed reduced levels of TNF-α and increased IL-10 levels, compared to untreated and TSG-6-depleted EV-treated. Moreover, we found that CD206 and Arg mRNA expression levels increased in the EV group but decreased in the TSG-6 depleted EV group (Figure 6A). To determine the effect TSG-6 contained in EV on macrophage polarization, the extent of M1 and M2 were confirmed. Accordingly, the number of CD206+cells increased in the EV group. However, in the TSG-6-depleted EV group, the proportion of CD206+cells decreased (Figure 6B). Contrary, the number of CD11c+cells decreased in the EV group compared to PBS group. However, in the TSG-6-depleted EV group, the proportion of CD11c+cells increased compared to EV group.

Figure 6. cAT-MSC-EV TSG-6 induced macrophage polarization from M1 to M2 type in vitro. LPS-stimulated canine macrophage (DH82) were cocultured for 48 h with cAT-MSC-EVs transfected with TSG-6 siRNA (si-TSG6) or scrambled siRNA (siCTL), or naïve EVs. (A) TNF-α, IL-10, CD206 and Arg mRNA-expression levels were measured (C) The M1 and M2 population were determined by measuring CD206 and CD11c positive cells by flow cytometry (n = 6 in each group). The results are presented as the mean ± standard deviation (*P < 0.05, **P < 0.01, ***P < 0.001, ****P < 0.0001, as determined by one-way ANOVA). 

TSG-6 in EV induced phenotypic enhancement of Tregs and M2 macrophage in inflamed colon

The FOXP3, CD4, CD25, CD206 and Arg mRNA levels were evaluated to examine whether stem cell EVs affected the activation of Tregs and polarization of macrophage in the inflamed colon. FOXP3 (6.8 fold), CD4(3.0 fold), CD25 (4.0 fold), CD206 (3.0 fold) and Arg (2.7 fold) mRNA levels increased in the EV group compared to the PBS group. However, their levels significantly decreased in the TSG-6-depleted EV group compared to the EV and CTL-EV group (Figure 7A). To determine whether the increase in the number of Tregs and M2 was associated with TSG-6 in the EVs, quantitative analysis of FOXP3+ cells and CD206+ cells detected in colon tissue sections by immunofluorescence showed that the percentage of FOXP+ cells (4.76 fold) and (7.61 fold) increased significantly in the cAT-MSC-EV group compared to that in the PBS group. However, the enhancement of the number of FOXP3+ cells (0.54 fold) and CD206+ cells (0.35 fold) in the EVs of the colon tissue decreased when TSG-6 was inhibited (Figure 7B).

Figure 7. TSG-6 increases regulatory T cells in the inflamed colon TSG-6 in EVs increased the proportion of Tregs in the inflamed colon. (A) Relative gene-expression levels of CD4, CD25, FOXP3, CD206 and Arg in the inflamed colon. (B) FOXP3+ (green) cells (C) CD206+ (gren)cells were detected in colon tissue sections by immunofluorescence. The data shown demonstrated that TSG-6 in EVs played a major role in increasing the number of Tregs and M2 in the colon. Scale bar, 50 μm. The results are shown as the mean ± standard deviation (n = 6–8 in each group, *P < 0.05, ****P < 0.0001, as determined by one-way ANOVA).”

3. In discussion section 

“Colon macrophages are essential for maintaining mucosal homeostasis for the ongoing need for epithelial regeneration, but are also an important component of protective immunity and are involved in the pathology of IBD (Bain and Mowat 2014). M1 preferentially metabolize arginine to nitric oxide via inducible nitric oxide synthase (iNOS; NOS2), while M2 preferentially metabolize arginine to ornithine via arginase -1. Therefore, it is known that M1 are involved in pre-inflammatory and M2 are involved in anti-inflammatory(Isidro and Appleyard 2016). In addition, The importance of macrophages in maintaining immune homeostasis has shown that IL-10 secreted from M2 acts on Treg to maintain Foxp3 expression(Murai et al. 2009). these Foxp3-expressed Tregs relieved inflammation by suppressing the activity of Th1 and Th17 cells(Xavier and Podolsky 2007). Therefore, increasing M2 in the colon is noted as a way to alleviate colitis (Song et al. 2018).”

COMMENT2-What is the cause of selecting TSG6 although proteins other than TSG-6, such as TGF-beta, IDO, PGE2 may also be contributing to the protective effect of EVs in relieving inflammation? A detailed justification may be required.

RESPONSE: In discussion section, we described this as follow. 

“Previous studies have reported that TGF-β plays a role in inhibiting activated immunity by inducing FoxP3+ regulatory t cells in an inflammatory environment and has been shown to play an important role in relieving inflammation in colitis models(Becker, Fantini and Neurath 2006). Zhang et al. reported that TGF-beta in bone marrow derived stem cells plays a major role in polarizing macrophage from M1 to M2 (Zhang et al. 2016) 

IDO expression and activity is an important mediator of intestinal homeostasis both in health and disease(Ciorba 2013). In addition, IDO appears to be the most promising candidate, which plays an important role in the immunomodulatory effects of stem cells by inhibiting T cell activation and enhancing Tregs(Yan et al. 2010). Also, IDO has been shown to play a major role in suppressing immunity by polarizing macrophage from M1 to M2.(Wang et al. 2014). Furthermore, IDO as an anti-inflammatory agent has been reported to reduce inflammation in colitis models(Coquerelle et al. 2009).

In our previous study, we confirmed the efficacy of adipose-derived stem cells in murine-derived macrophage cell lines in inflammatory environments and demonstrated that PGE2 secreted from stem cells is a key factor in polarizing macrophage (Yang et al. 2018, Chae et al. 2017). Moreover, we previous showed that PGE2 secreted feline ASC is a key factor for enhancing regulatory T cell in inflamed colon(AN et al. 2018)

Although further research on the correlation between these immunomodulatory factors of EV and immune cell regulation is needed, results of the current study confirm that TSG-6-depleted EVs significantly reduce the immunoregulatory ability, which clearly indicates that TSG-6 is a major factor in immune regulation and anti-inflammatory action. Furthermore, the finding that TSG-6 in EVs plays an important role in immune regulation will serve as evidence to support increasing the level of TSG-6 in EVs as a strategy to develop EVs with enhanced immunomodulating properties.

We hope you agree.

COMMENT3-Increasing the level of TSG-6 in EVs as a strategy to develop EVs with enhanced immunomodulating properties is thought to be a very promising idea that needs to be addressed as a major recommendation or even to be added in the conclusion section.

RESPONSE: Thank you for the good comment. We described this in conclusion section as follow

“ We demonstrated that TSG-6 in EVs secreted from cASCs ameliorated DSS-induced colitis in mice by enhancing the Treg population and polarizing macrophage from M1 to M2 in the inflamed colon. Our findings provide an insight to improve the current understanding of the role that EVs have in immunoregulation and serve as a foundation for applying EVs as a therapeutic agent in IBD. Also, this study is the basis of a strategy for developing EVs with improved immunomodulatory properties by increasing TSG-6 levels in EVs.”

COMMENT 4 The abbreviation TEM (page 14, line 17)is not included in the abbreviations list.

RESPONSE: Thank you for the good comment. We described this in the abbreviations list. as follow 

“TEM: Transmission Electron Microscope”

COMMENT 5 Page 13, line 9; as previously described not as previous.

RESPONSE: Thank you for the good comment. We described this in the M&M section. as follow

“Colon sections were deparaffinized and rehydrated, and antigen retrieval was carried out in 10 mM citrate buffer. Sections were then washed and blocked with blocking buffer containing 1 % bovine serum albumin and 0.1 % tween 20 for 30 min. The sections were then incubated overnight at 4℃ with mouse monoclonal anti-Forkhead box (Fox) P3 (1:100; Santa Cruz Biotechnology) and mouse monoclonal anti CD206 (1:100; Santa Cruz Biotechnology). The colon sections were washed three times with DPBS. Then, the sections were incubated FITC conjugated anti mouse (1:500; Santa Cruz) for 1 hr. After that, they were washed three times with DPBS. All samples were mounted using Vectashield mounting medium containing 4’,6-diamidino-2-phenylindole (DAPI; Vector Laboratories, Burlingame, CA, USA). The samples were observed using an EVOS FL microscope (Life Technologies, Darmstadt, Germany). Immunoreactive cells were counted in 20 random fields per group, and the percentage of CD206+ positive cells and FOXP3+ positive cells was calculated in colon sections from the same mice.” 

COMMENT 6 Page 24, line 12; immune mediated diseases not disease.

REPONSE: Thank you for the good comment. This paragraph has been removed from the manuscript.

 

Reviewer #2: 

COMMENT 1: you write in the experimental design that is 6 groups but in some figure 5 group only

RESPONSE: Thank you for the good comment. We created a sham model to verify that EV had no side effects even when injected into normal mice. As a result, mice injected with EV showed no difference in body weight, activity, and stool consistency from normal mice. However, in the DSS-induced mice, the DAI was increased, the stools were soft, and histological examination also confirmed that there were a lot of inflammatory cells. The next step was to determine if EV had an anti-inflammatory effect in mice induced with DSS through immunomodulation. Therefore, we focused on the case of injection of EV into DSS mouse and no injection, and also injection of EV with reduced TSG-6. In this regard, we described in results section. 

“Injecting EV into mice that did not induce colitis (sham group) showed no difference in vitality, weight, and stool consistency from naive mice. In addition, the sham group was not different from the naive group in the colon length and histological examination.”

COMMNET 2: CTl , si-tsg-6 are not present in the abbreviation list and not mentioned in experimental design

RESPONSE: Thank you for the good comment. We described this in abbreviation section and M&M section. 

In abbreviation section 

“siTSG-6: Small-interfering TSG-6, CTL: Control”

In M&M section 

“To obtained TSG-6 depleted EV, when cASCs reached approximately 70 % confluence, they were transfected for 48 h with TSG-6 siRNA or control siRNA (sc-39819 and sc-27007, respectively, Santa Cruz Biotechnology, Dallas, TX, USA) using Lipofectamine RNAiMAX (Invitrogen, Carlsbad, CA, USA) according to the manufacturer’s instructions”

COMMENT3: what is the difference between CTL-MSc-EV and TSG 6 depleted EV. what is the significance of using these two groups?

RESPONSE: Thank you for the good comment. 

In this experiment, stem cells were treated with siTSG-6 to reduce TSG-6 in EV. Therefore, stem cells were transfected for 48 h with TSG-6 siRNA or control siRNA (sc-39819 and sc-27007, respectively, Santa Cruz Biotechnology, Dallas, TX, USA) using Lipofectamine RNAiMAX (Invitrogen, Carlsbad, CA, USA) according to the manufacturer’s instructions. In order to create a control group for genetically modified with siTSG-6 in ASC, an EV group obtained from siRNA ASC was created. Therefore, I think it is important to compare siRNA (control group (CTL)) with TSG-6 depleted EV group. For this reason, the CTL group and TSG-6 depleted EV group were compared statistically in this study. 

COMMENT4: experimental design needs to clarify (page no 12)

RESPONSE: Thank you for the good comment. We described this in M&M section. 

“To determine the therapeutic effect of ASC-derived EVs on colitis, we induced mouse colitis with DSS (36–50 kDa; MP Biomedical, Solon, OH, USA).”

COMMENT 5: there is very high S.D in the results such as figure 4, 5 and 6.

why you didn't use DSS 5%

RESPONSE: Thank you for the good comment. 

The mouse used in this experiment is BL6, and the recommended dose ranges from 1.5% to 3.0% (Benoit Chassaing et al. Curr Protoc Immunol, 2014). Therefore, we used that concentration to induce colitis. And it was found that the colon tissues were significantly inflamed compared to the Naive group. Moreover, in our previous study, also we used 3% DSS to induce colitis in mice. In other previous study, since the incidence of colitis in DSS depends on the immunity and lifestyle of the mouse, the deviation may be different even if DSS is fed (Benoit Chassaing et al. Curr Protoc Immunol, 2014).

COMMENT 6: why you didn't analyze the stool EV, colon length

RESPONSE: Thank you for the good comment. We analyzed colon length, and this were described in results section as follow. 

“Moreover, shortening of the colon length significantly improved in the EV group compared with that in the PBS-treated group. However, in TSG-6 depleted EV group was found to have a shorter colon length than the EV and CTL-EV groups”

And stool consistency was analyzed in DAI scoring. These are described in results section.

COMMNET 7: also why you didn't measure NF-KB, p65, INOs, caspase 3 is a relation between TSG 6 and TSG 14 in colitis

RESPONSE: Thank you for the good comment. This study aimed to see if the TSG-6 contained in the EV might play a role in alleviating colitis. The study also focused on the relationship between TSG-6 and regulatory t cells and macrophage M2 type. 

In discussion section, this were descried as follow. 

“ Nuclear transcription factor kappaB (NF-κB) is a central mediator of pro-inflammatory gene induction and function in immune cells and has a significant effect on mucosal inflammatory process(Liu et al. 2017). Moreover, in IBD patients, its activation is markedly induced. Therefore, the NF-κB pathway is considered to be an attractive target of therapeutic intervention in IBD(Atreya, Atreya and Neurath 2008). In our previous study, TSG-6 from stem cells significantly suppressed nuclear factor kappa B (NF-κB) activity and alleviated inflammation and reduced apoptosis in acute pancreatitis model(Li et al. 2018). However, the relationship between EV and NF-kB has not been studied in this study, and it is necessary to confirm whether EV's TSG-6 lowers NF-kB in inflammatory colon.”

COMMENT 8. in reference No 9 (the paper on renal not on colon) you need to check the paper

https://www.ncbi.nlm.nih.gov/pmc/articles/PMC3303802/

RESPONSE: Thank you for the good comment. We revised this in revised manuscripts. 

In introduction section 

“Recently, various studies have been carried out on the application of EVs as therapeutic agents in various pre-clinical models such as acute kidney injury, hepatitis, cystitis and uveitis(Hartjes et al. 2019, Qian et al. 2016, Bai et al. 2017, Bruno et al. 2012). In addition, injecting EVs into DSS-induced colitis mouse models has shown that not only does it improve activity and appetite, but it also alleviates inflammation in the colon (Mao et al. 2017). Although these studies have reported that damaged tissues were improved following treatment with EVs, the factors responsible for the protective effects have yet to be elucidated.”

Again, we appreciate all of your insightful comments. We worked hard to respond to them. Thank you for taking the time and energy to help us improve this manuscript.

Sincerely yours,

Hwa-Young Youn, D.V.M., Ph.D.

Professor

Department of Veterinary Internal Medicine, College of Veterinary Medicine, 

Seoul National University, Seoul 08826, Republic of Korea

Tel : +82-2-880-1266

E-mail : hyyoun@snu.ac.kr

Woo-Jin Song, D.V.M., Ph.D.

Clinical Assistant Professor,

Laboratory of Veterinary Internal Medicine, Department of Clinical Science,

College of Veterinary Medicine, Seoul National University,

Seoul 08826, Republic of Korea

Tel : +82-2-880-8661

E-mail : woojin1988@snu.ac.kr

---

## [Editor Report · Decision Letter 1]

9 Dec 2019

TSG-6 in extracellular vesicles from canine mesenchymal stem/stromal is a major factor in relieving DSS-induced colitis

PONE-D-19-20533R1

Dear Dr. Youn,

We are pleased to inform you that your manuscript has been judged scientifically suitable for publication and will be formally accepted for publication once it complies with all outstanding technical requirements.

With kind regards,

Hossam MM Arafa

Academic Editor

PLOS ONE
---

## [Editor Report · Acceptance letter]

28 Jan 2020

PONE-D-19-20533R1 

TSG-6 in extracellular vesicles from canine mesenchymal stem/stromal is a major factor in relieving DSS-induced colitis 

Dear Dr. Youn:

I am pleased to inform you that your manuscript has been deemed suitable for publication in PLOS ONE. Congratulations! Your manuscript is now with our production department. 

With kind regards,

on behalf of

Professor Hossam MM Arafa 

Academic Editor

PLOS ONE